# Strengthening the Link between Vaccine Predispositions and Vaccine Advocacy through Certainty

**DOI:** 10.3390/vaccines10111970

**Published:** 2022-11-20

**Authors:** Borja Paredes, Miguel A. Martín Cárdaba, Lorena Moreno, Nerea Cano, Pablo Briñol, Ubaldo Cuesta, Richard E. Petty

**Affiliations:** 1Departamento de Psicología Social y Metodología, Facultad de Psicología, Universidad Autónoma de Madrid, 28049 Madrid, Spain; 2Facultad de Comunicación, Universidad Villanueva, 28034 Madrid, Spain; 3Departamento de Teoría y Análisis de la Comunicación, Facultad de Ciencias de la Información, Universidad Complutense de Madrid, 28040 Madrid, Spain; 4Department of Psychology, Ohio State University, Columbus, OH 43210, USA

**Keywords:** beliefs about medicines questionnaire (BMQ), vaccination attitudes examination scale (VAX), vaccination advocacy, individual differences, meta-cognition, certainty, confidence

## Abstract

Background. Instruments designed to assess individual differences in predispositions towards vaccination are useful in predicting vaccination-related outcomes. Despite their importance, there is relatively little evidence regarding the conditions under which these instruments are more predictive. The current research was designed to improve the ability of these kinds of instruments to predict vaccination advocacy by considering the certainty associated with the responses to vaccination scales. Method. Across two studies, participants completed the Beliefs about Medicines Questionnaire BMQ scale (Study 1) or the Vaccination Attitudes Examination (VAX) scale (Study 2). The certainty participants had in their responses to each scale was either measured (Study 1) or manipulated (Study 2). Intentions to advocate in favor of vaccination served as the criterion measure in both studies. Results. As expected, the scales significantly predicted vaccination advocacy, contributing to enhancing the predictive validity of the instruments used in the studies. Most relevant, certainty moderated the extent to which these scales predicted vaccination advocacy, with greater consistency between the initial scores and the subsequent advocacy willingness obtained for those with higher certainty. Conclusions. Certainty can be useful to predict when the relationship between vaccination-related cognitions (i.e., beliefs or attitudes) and advocacy willingness is likely to be stronger.

## 1. Introduction

Vaccines are one of the most effective medical tools in preventing disease, disability, and death [1]. As the World Health Organization (WHO) suggests, vaccination prevents 4–5 million deaths every year [2]. Additionally, in the context of the pandemic caused by COVID-19, a recent study estimates that 470,000 lives have been saved among those aged 60 years and above since the start of the COVID-19 vaccination roll-out in 33 countries across the WHO European Region [3].

Yet, despite all their potential benefits, there is variance in the extent to which people believe vaccines are useful, in the extent to which people favor their usage, and advocate for them [4,5,6]. This variation in the level of advocacy is important because most vaccines require having a high degree of social acceptance to work at the community level (i.e., preserving herd immunity and preventing outbreaks of vaccine preventable diseases) [7]. We define advocacy as a person’s stated or actual willingness to share one’s opinion on a topic with others [8,9,10]. Advocacy is sometimes referred to as word of mouth [11], dissemination [12] or forwarding [13]. Advocacy of vaccination is also important for the adoption of novel vaccines as in the current case of COVID-19 [14]. Additionally, in countries where certain political parties have not advocated in favor of getting vaccinated, supporters of such parties have significantly lower vaccination rates [15]. This is one example in which vaccination advocacy could be affecting vaccine hesitancy, defined as the delay in uptake or refusal of a vaccine even when the vaccine is proven safe and is widely available. Hesitancy is among the top ten global health threats identified by the World Health Organization [16]. Therefore, it is important to recognize vaccine hesitancy and address it early before it leads to vaccine refusal.

Understanding beliefs and attitudes underlying vaccination is important for predicting relevant outcomes in this domain [17,18,19]. Therefore, different questionnaires have been developed and validated to measure people’s attitudes towards vaccines such as the Beliefs about Medicines Questionnaire (BMQ) [20] and the Vaccination Attitude Examination (VAX) Scale [21]. More examples of these measurements exist, such as the vaccine hesitancy scale [22], and the knowledge of vaccination scale [23]. For this research, we chose the scales about beliefs and attitudes because they are relatively less studied, and because they have been recently adapted to the target Spanish population studied. The main purpose of the present research is to examine the ability of these instruments to predict relevant outcomes, and to increase the predictive validity of vaccination-related scales by taking into consideration the certainty that people have in their responses to such scales. Specifically, we aim to measure certainty in an initial exploratory study to test the presence of such effect, and then we aim to manipulate certainty in a follow-up study to examine a causal inference of this effect.

There is expanding literature which suggests that to better predict behavior from mental constructs, one should also consider the certainty with which the relevant mental construct is held. Mental constructs are better predictors of intentions and behavior when people hold the mental constructs with higher certainty. For example, belief certainty can moderate the correspondence between beliefs and attitudes as illustrated by research on the self-validation theory (SVT) [24]. Furthermore, attitude certainty has been shown to qualify the correspondence between attitudes and subsequent behavior [25,26,27]. Certainty refers to the subjective sense of conviction or confidence that people have in their mental content, or the extent to which they believe their mental content is correct or valid [28].

Beyond beliefs and attitudes, the recent literature indicates that certainty can moderate the predictive validity of individual-difference scales. For example, Shoots-Reinhard et al. [29] demonstrated that individual differences in the need to evaluate scale [30] and in political ideology [31] became better predictors of relevant outcomes in their respective domains as individuals’ certainty in their scores to those instruments increased (for additional examples [32,33,34,35,36,37,38]). In sum, prior research has revealed that certainty can moderate the reliance on mental constructs, including those assessed with validated instruments in various domains. The current set of studies examined for the first time whether certainty in one’s responses to vaccination-related scales can also moderate the relationship between these vaccine inventories and vaccine-related outcomes.

Unlike relatively intuitive frameworks for which certainty is seen as a precursor only of positive outcomes (e.g., recovery [39]), we propose that certainty has the potential to magnify the effect of any mental construct including both positive and also negative predispositions toward vaccines.

The primary goal of the present research is to contribute to the predictive validity of vaccination-related scales [20,21]. Most relevant, the present studies were designed to provide an efficient strategy to strengthen the relationship between these scales and vaccination-related outcomes by taking certainty into consideration.

Study 1 examined to what extent measuring certainty in the BMQ can contribute to specifying when the relationship between the BMQ and vaccine advocacy willingness will be stronger. Study 2 was designed to extend this effect by using a different vaccination-related scale (VAX, [21]) and by manipulating certainty rather than measuring it. As noted, we predicted that certainty would moderate the ability of both scales to predict advocacy willingness across studies. Given that COVID-19 vaccines were already widely available at the time of data collection, we decided to focus on another relevant outcome regarding vaccination (i.e., willingness to share messages). The greater the certainty (regardless of measured or manipulated), the stronger we expected the link between the scales and subsequent intentions to advocate in favor of vaccines. The core idea is that before advocating for vaccines or deciding to advocate, people are likely to consider not only their opinion toward vaccines but also how certain they are in their view. The more certain they are in their vaccine beliefs, the more likely they are to act in accord with them.

## 2. Study 1

The goal of this first study was to (1) provide convergent evidence of the link between a vaccination-related scale and advocacy willingness, and (2) test the more novel prediction that certainty in the scale responses would moderate the predictive validity of the scale. First, participants responded to the Beliefs about Medicines Questionnaire (BMQ). We decided to use a more ‘general’ scale that could relate to several medically related outcomes, vaccination advocacy included. Participants then reported their degree of certainty in their responses to the scale. These two variables and their two-way interaction served as predictors of advocacy willingness (i.e., the criterion measure). As noted, advocacy is an important outcome in this context for several reasons. First, advocacy has been associated with other relevant behaviors such as those that are consistent with the advocated position [40]. Furthermore, research has shown that the more people are persuaded as a result of that advocacy, the stronger the social consensus about vaccination [41,42]. Second, advocacy has also been found to be critical for convincing oneself (i.e., self-persuasion) and others [43]. Third, recent research has found that the more people advocate for a cause such as vaccines, the less likely it is for fake news to be accepted without criticism [44]. Consistent with recent research in this domain [45], we expected the BMQ to be associated with advocacy willingness. Most relevant to the present purposes, we also expected that, as participants’ certainty in their answers to the BMQ increased, so too would the relationship between such responses and participants’ advocacy willingness.

### 2.1. Materials and Methods

One hundred and eighty-six undergraduate students (31.7% males) from Universidad Complutense de Madrid participated anonymously in this study. All participants were recruited in exchange for extra credit in one of their courses. The age of the participants ranged from 18 to 57 (M_age_ = 22.98, SD = 6.56). The study was carried out in a laboratory room at the university where participants were asked to complete a survey on a computer using Qualtrics software. A power analysis was conducted using G*Power [46]. We could not look at prior work to obtain an estimated effect size for the predicted interaction between vaccinations attitudes and certainty. Therefore, we planned for a generic relatively small effect (Cohen’s f^2^ = 0.050). Results indicated that the desired sample size (α = 0.05) with these parameters at 0.80 power was N = 159 participants. We decided to keep collecting participants until the end of day in which the study was conducted in case there were unexpected missing values, resulting in a final sample of N = 186. The two-way interaction was also significant when the analysis was run using the first 159 participants like the G*Power analysis suggested (*B* = 0.320, *t*(155) = 3.110, *p* = 0.002, 95% CI: [0.117, 0.524]).

Participants were provided with a brief passage that described the experiment as a study on scale validation. Participants first completed the Beliefs about Medicines Questionnaire [20]. Next, participants reported their certainty in their responses to the scale. Lastly, participants were asked about their intentions to spread messages on social networks in favor of vaccination, which served as the dependent variable in this first study (i.e., advocacy willingness). After responding to these questions, participants were then debriefed, thanked and dismissed.

### 2.2. Predictor Variables

*Beliefs about Medicines Questionnaire (BMQ):* This scale assesses the beliefs people hold about medications, their known and unknown side effects, and potential adherence to prescribed treatments [20]. The responses to these items ranged from 1 “totally disagree” to 5 “totally agree” (α = 0.80). Responses were scored so that higher numbers represented more positive beliefs about medicines (M = 3.45, SD = 0.66). The revised model of the European Federation of Psychologists’ Association (EFPA) for the evaluation of the quality of tests [47] and the established recommendations for successfully adapting measures from one culture to another were followed to ensure the accuracy of the final translation [48]. The BMQ has shown to be able to predict medication-related behavior, including willingness to receive vaccines [45] and adherence to medical treatment when prescribed [20]. In order to test the factorial validity of the Spanish scale compared to the original, an exploratory factor analysis was run with the sample used in Study 1 (N = 186), using the Pearson correlation matrix, maximum likelihood as the estimation method, and direct oblimin as the rotation method [49]. Based on the original article [20], two factors were extracted, explaining more than 57 percent of the total variance. The two factors extracted in the original article (i.e., Harm and Overuse) also explained a similar amount of variance (i.e., 51 percent). The scale is comprised of two subscales, namely, “Overuse” (α = 0.66) and “Harm” (α = 0.66). Examples of the items of this instrument include “Doctors prescribe too much medication” (from the “Overuse” subscale) and “Medical treatments do more harm than good” (from the “Harm” subscale).

*Certainty:* Following completion of the scales, participants were asked to think back to the BMQ scale and report the confidence they had in their responses. Participants provided their certainty ratings on three items, including certainty, confidence, and validity about their responses to the BMQ scale. Responses to these three items were measured on 7-point scales (1 = not at all certain/none at all confident/ not at all valid to 7 = very certain/very confident/ extremely valid, respectively). A composite index of certainty was formed by averaging responses to these three measures (α = 0.73). Higher values on this index indicated more certainty in participants’ responses to the BMQ (M = 5.67, SD = 0.83).

### 2.3. Dependent Variable

*Advocacy Willingness:* Participants were asked to report their intentions towards spreading messages in favor of vaccination using two 7-point Likert scales ranging from 1 “totally willing to” to 7 “totally unwilling to”. We asked participants two questions: “To what extent are you willing to share messages on social networks in favor of vaccines?” and “To what extent are you willing to share messages on social networks in favor of the COVID-19 vaccine? (M = 6.58, SD = 0.91). Responses were coded so that higher numbers represented greater intentions to spread messages supporting vaccination (r = 0.84). Previous research in the realm of vaccination focused on reported intentions to receive the COVID-19 vaccine “when it was made available” [46]. As noted, we decided to adapt the questions given that COVID-19 vaccines were already widely available at the time of data collection for this study by focusing on willingness to share messages. Additionally, relevant prior research has measured willingness to share messages on social media using similar items [50]. A pilot study was conducted to provide empirical support for the assumption that willingness to advocate in favor of vaccination can predict actual behavior. In this study, 66 participants completed the same two items about advocacy willingness that we used in this study and were asked to sign a petition to support the development and rollout of vaccines of medical and pharmaceutical associations (2 = Signed the petition, 1 = did not sign the petition). A binary logistic regression was conducted using advocacy intentions as the predictor, and the behavior of signing the petition as the dependent variable. As expected, advocacy intentions were found to be a significant predictor of actual behavior (*B* = 0.453, z = 7.398, SE = 0.167, *p* = 0.007), such that intentions to advocate in favor of vaccination were positively associated with signing the petition to support the development of vaccines. Specifically, looking at the odds ratio value [Exp(B) = 1.573], as participants increased 1 point in the advocacy willingness score, their odds of signing the petition in support of vaccination increased by 57%.

As noted, intentions to spread online messages matter because of their potential real-life consequences [51,52]. Advocacy intentions have been associated not only with proselytism [53] but also with negative evaluations and stigmatization of individuals against the advocated position [54] and aggressive defense of the cause [55]. In addition, the relationship between different forms of advocacy intentions and subsequent behavior has been replicated in different fields of social sciences, namely, consumer behavior [56,57] and vaccination research itself [58].

### 2.4. Data Analysis

Following the suggestions of Cohen and Cohen [59], advocacy willingness was submitted to a hierarchical regression analysis. The VAX scale (centered), certainty (centered), and the interaction term were entered as predictors. Main effects were interpreted in the first step of the regression and the two-way interaction in the second, final step. The critical two-way interaction was tested using the PROCESS add-on for SPSS (model 1).

### 2.5. Results

The results indicated a main effect of the BMQ scale (*B* = 0.374, *t*(182) = 3.851, *p* < 0.001, 95% CI: [0.182, 0.566]), indicating that people scoring higher in the BMQ scale are more willing to spread pro-vaccine messages on social media. The main effect of certainty did not reach statistical significance (*B* = 0.009, *t*(182) = 0.127, *p* = 0.898, 95% CI: [−0.140, 0.159]). The predicted two-way interaction between the BMQ scale and certainty was significant, (*B* = 0.304, *t*(182) = 2.949, *p* = 0.003, 95% CI: [0.100, 0.507]). When using each of the two sub-scales as predictors instead of the entire BMQ scale (Harm and Overuse), the two-way interaction remained significant for both sub-scales, (Harm; *B* = 0.302, *t*(182) = 2.994, *p* = 0.003, Overuse; *B* = 0.211, *t*(182) = 2.248, *p* = 0.025).

As illustrated in Figure 1, among those with relatively higher certainty scores (+1SD), the BMQ scale was positively associated with advocacy willingness (*B* = 0.629, *t*(182) = 5.011, *p* < 0.001, 95% CI: [0.381, 0.877]). However, for those with lower certainty scores (−1SD), the relationship between BMQ and advocacy willingness was not significant (*B* = −0.119, *t*(182) = 0.889, *p* = 0.374, 95% CI: [−0.145, 0.385]).

### 2.6. Discussion

As expected, the BMQ scale predicted intentions to advocate in favor of vaccines via social networks, providing convergent evidence of the predictive validity of the BMQ scale. Most relevant for the present concerns, the effect of the BMQ on advocacy willingness was moderated by reported certainty in the scale responses. As hypothesized, we found that the BMQ was associated with advocacy willingness to a greater extent as participants were more certain in their responses. Thus, as participants’ certainty in their responses increased, so too did the ability of the BMQ to predict the intentions related to the dissemination of messages in favor of vaccines through social networks. This suggests that researchers interested in using the BMQ scale can benefit by adding an additional measure of certainty.

Given that certainty in the first study was measured rather than manipulated, there exists the possibility of reverse causality (i.e., that instead of certainty increasing the relation between BMQ and vaccination advocacy, a high relation between BMQ and vaccination advocacy leads people to infer certainty). Lastly, one might argue that there is a relatively low level of connection between beliefs in medicines in general (i.e., BMQ) and willingness to advocate in favor of vaccines in particular. Although the scale was capable of predicting the intended outcomes, the level of specificity between measures might be seen as relatively low.

In order to address these issues and isolate the causal effect of certainty, as well as to generalize our results even further, we moved in the next study towards an experimental design by manipulating certainty. Furthermore, study 2 used a scale specifically matching attitudes towards vaccinations.

## 3. Study 2

In this study, participants were first asked to complete the Vaccination Attitudes Examination (VAX) scale [21]. Earlier studies have found that the VAX scale is significantly associated with actual vaccination (e.g., influenza vaccination and COVID-19), as well as intentions to receive future vaccines either for oneself or for one’s children [21,60]. We aimed at generalizing our findings by using a specific vaccination attitudes scale as our main predictor. After responding to the VAX scale, participants were randomly assigned to the certainty or doubt condition. Participants were assigned to write a personal experience in which they felt either confidence or doubt. Finally, participants were asked to report their advocacy willingness regarding vaccinations, the same instrument used in the first study with one additional item to increase reliability. It was expected that the VAX scale would be significantly more predictive of advocacy willingness in the certainty (vs. doubt) condition.

### 3.1. Materials and Methods

Two hundred and two undergraduate students (32.1% males) from Universidad Complutense de Madrid participated anonymously in this study. All participants were recruited in exchange for extra credit in one of their courses. The age of the participants ranged from 18 to 57 (M_age_ = 22.87, SD = 6.46). We kept collecting participants during the rest of the academic semester, and then we ran a sensitivity analysis using G*Power [46] for a two-tailed test (α = 0.05) of the predicted 2-way interaction with 0.80 power. We aimed for a sample size big enough to detect the effect in Study 1 (Cohen’s f^2^ = 0.043). The sample size we collected (N = 202) was able to detect effects larger than f^2^ = 0.039.

Participants were provided with a brief passage describing the experiment as a study on scale validation. Participants first completed the Vaccination Attitudes Examination (VAX) Scale. Next, participants were randomly assigned to describe in detail a recent personal experience in which they felt either confidence or doubt about their thoughts. Lastly, participants were asked about their intentions to spread messages on social networks in favor of vaccination, which served as a dependent variable. After responding to the questions, participants were then debriefed, thanked and dismissed.

### 3.2. Predictor Variables

*Vaccination Attitudes Examination (VAX):* The VAX scale is one of the most commonly used scales for understanding attitudes towards vaccines. Recently validated to different languages [45,61,62,63,64], this instrument consists of a 12-item questionnaire formed by four specific subscales that evaluate: (a) mistrust of vaccine benefit, (b) worries about unforeseen future effects, (c) concerns about commercial profiteering, and (d) preference for natural immunity. Unlike other scales, the VAX scale does not focus exclusively on specific vaccines and higher total VAX scores suggest stronger anti-vaccination attitudes. As noted, the VAX scale has shown to be a significant predictor of several vaccine-related intentions and behaviors [21,45,60]. The scale was coded so that higher numbers indicated more favorable vaccination attitudes (M = 4.33, SD = 0.7), indicating their agreement with each item (1 = Totally Disagree, 6 = Totally Agree). Examples of items include: “I feel safe after being vaccinated” and “I worry about the unknown effects of vaccines in the future” (α = 0.85). The VAX has shown to be able to predict willingness to receive vaccines [45]. Similar to its original version [21], an exploratory factor analysis was run in Study 2’s sample (N = 202), using the Pearson correlation matrix, maximum likelihood as the estimation method, and direct oblimin as the rotation method [49]. Like the original article and the Spanish validation article, its results showed the current version to be compatible with a four-dimensional factorial structure (i.e., explaining 69 percent of the variance).

*Certainty:* Following completion of the scales, participants were assigned to write a past personal experience in which they felt either confidence or doubt. Some examples of experiences listed in the doubt condition included: “When I moved to another country, I started to doubt if I had made the right decision because it was a big change in my life”. Some examples of experiences recalled in the confidence condition were: “When the other day I was arguing with my family about public education”. Previous research has shown that this manipulation works to induce metacognitive certainty and doubt [33,65]. Self-validation theory (SVT) highlights that variables completely incidental to the accessible mental content can have an impact on their perceived validity [24]. The logic behind this manipulation is that recalling past episodes of confidence gives people a momentary feeling of confidence that can be then misattributed to whatever is in mind at the time (in this case, the responses to the scale). Although it was unlikely that this manipulation affected the scores of the scale (since it followed the completion of the inventory), nonetheless, we still examined this relationship and confirmed that there was no effect of the induction on the measure of VAX, *p* = 0.931.

### 3.3. Dependent Variable

*Advocacy Willingness:* Participants were asked to report their intentions towards supporting vaccines using the same two 7-point Likert scales from Study 1 (α = 0.88) plus the following one: To what extent are you willing to sign a petition defending the widespread use of vaccines?” (α = 0.88; M = 4.61, SD = 1.65). The two-way interaction was significant when only the two items from Study 1 were used as a dependent variable (*B* = 0.281, *t*(197) = 2.03, *p* = 0.043). The two-way interaction was also significant when the new item alone was used as a dependent variable (*B* = 0.311, *t*(197) = 2.27, *p* = 0.024). This measure was included due to its similarity to the outcome measure used in the pilot study, in addition to being used in previous research about word-of-mouth through social media [66,67] and in previous research regarding vaccination intentions [46].

### 3.4. Data Analysis

Following the suggestions of Cohen and Cohen [59], advocacy willingness was submitted to a hierarchical regression analysis. The VAX scale (centered), certainty (centered), and the interaction term were entered as predictors. Main effects were interpreted in the first step of the regression and the two-way interaction in the second, final step. The critical two-way interaction was tested using the PROCESS add-on for SPSS (model 1). The continuous variable (i.e., the VAX scale) was mean-centered.

### 3.5. Results

The regression analysis indicated a main effect of the VAX scale on advocacy willingness (*B* = 0.958, *t*(198) = 6.606, *p* < 0.001, 95% CI: [0.672, 1.245]), indicating that people scoring higher on the VAX scale were more willing to promote vaccination. The main effect of certainty did not reach statistical significance (*B* = 0.143, *t*(198) = 1.367, *p* = 0.173, 95% CI: [−0.063, 0.350]). More central to our predictions, the interaction between the VAX scale and certainty was significant (*B* = 0.323, *t*(198) = 2.227, *p* = 0.027, 95% CI: [0.037, 0.610]). As illustrated in Figure 2, among those assigned to the certainty condition, the VAX scale was associated with higher advocacy willingness (*B* = 1.256, *t*(198) = 6.273, *p* < 0.001, 95% CI: [0.861, 1.651]). Additionally, for those assigned to the doubt condition, there was a significant relationship between the VAX scale and advocacy willingness, although significantly weaker (*B* = 0.61, *t*(198) = 2.898, *p* = 0.004, 95% CI: [0.195, 1.024]). When using each of the four VAX subscales ((1) mistrust of vaccine benefit, (2) worries about unforeseen future effects, (3) concerns about commercial profiteering, and (4) preference for natural immunity) as predictors instead of the entire scale, the two-way interaction only remained significant in one of the four (i.e., concerns about commercial profiteering) (*B* = 0.242, *t*(198) = 2.032, *p* = 0.043, 95% CI: [0.007, 0.476]).

### 3.6. Discussion

Besides the changes introduced in this second study, we still found that another instrument related to vaccines (the VAX scale) predicted advocacy willingness. Participants’ responses to the scale were more associated with advocacy willingness in the certainty (vs. doubt) conditions. By manipulating certainty, this study offers evidence in favor of a causal role of certainty in moderating the relationship between responses to the scale and advocacy willingness.

## 4. Conclusions

The current research provided convergent evidence reaffirming the predictive validity of the BMQ and VAX scales by showing their capability of predicting relevant vaccination-related outcomes. Most importantly, across two studies, the results support our hypothesis that certainty moderates the effects of vaccination-related attitudes on relevant behavioral intentions. Specifically, we found that responses to the scales were associated with advocacy willingness to a greater extent when participants were more certain of their responses to the scale (both measured and manipulated). Thus, as certainty increased, so too did the association between these scales and subsequent advocacy willingness. These findings show that in order to maximize the predictive validity of the scales, it would be advisable for these scales to be accompanied by an extra measure of the extent to which participants feel certain in their answers. Furthermore, these results could help to explain the existence of individuals who, despite expressing favorable attitudes towards vaccines, do not translate these beliefs into pro-vaccination behaviors. Thus, this new insight could be used to identify target audiences for specific interventions aimed at increasing the certainty of their beliefs.

Having shown the predicted effects on advocacy willingness measures, several avenues for future research may come up. First, future research could benefit from studying the impact of certainty not only on self-reported measures of advocacy willingness but in actual behavior. In addition, when it comes to self-reported vaccination-related behavior, further clarification is needed regarding the extent to which the scale is predicting more or less accurately when certainty is high (e.g., confidence increasing accuracy in recalling prior responses of past behavior), or whether higher certainty coming right after the scale is causing them to report subsequent behavior that is more in line with their self-reports (i.e., I am certain that I am in favor of vaccines and I should report being willing to advocate to get vaccinated). Future studies can benefit from specifying whether and when motivations for accuracy or motivations for consistency are likely to be involved.

Future endeavors should explore whether the moderating role of certainty in the association between individual-difference scales and behavior can be extended to examine other instruments originally developed to assess other forms of treatment (e.g., psychological therapy). Furthermore, future research can benefit from examining the interaction between this scale and other scales designed to capture individual differences in the propensity to hold mental constructs with certainty [68], and with other scales that increase the reliance on confidence [69]. One additional aspect about this research is that the general relationship between attitudes and intentions in this particular topic might have been lower than usual, given the overall level of uncertainty at the time of the pandemic in which the studies were carried out.

Finally, future research should also specify the conditions under which perceived validity of vaccination attitudes is more likely to be taken into consideration. Self-validation theory [24] postulates that reliance on metacognitive assessments, such as certainty, is more likely to occur for individuals motivated and able to engage in thinking. Specifically, our experimental sample was comprised of undergraduate university students that were expected to have higher academic knowledge and more familiarization with the scientific field. In line with the self-validation theory [SVT; 24], these moderating effects of certainty might take place, especially for people who are involved enough in the topic to consider not only their attitudes about such a topic, but the certainty with which they hold their attitudes. Other less involved samples might have a more heuristic use of certainty. In line with multiple roles of variables in persuasion settings [70], future research should study the extent to which certainty has different roles in predicting the association between attitudes and intentions as a function of involvement.

## Figures and Tables

**Figure 1 vaccines-10-01970-f001:**
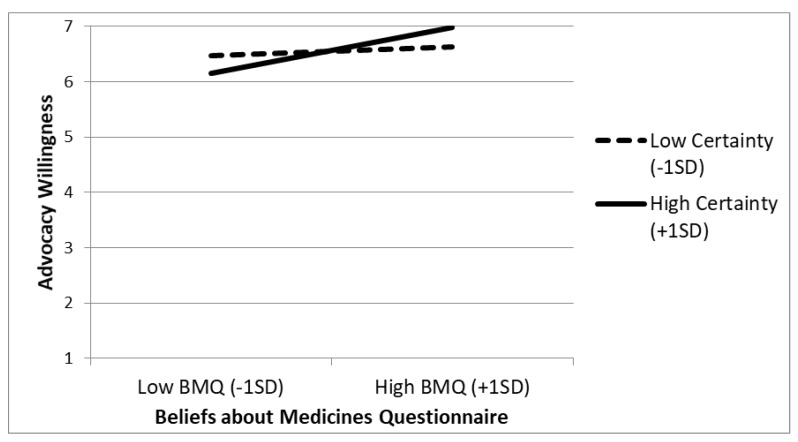
Study 1. Advocacy Willingness as a Function of BMQ Responses and Reported Meta-Cognitive Certainty.

**Figure 2 vaccines-10-01970-f002:**
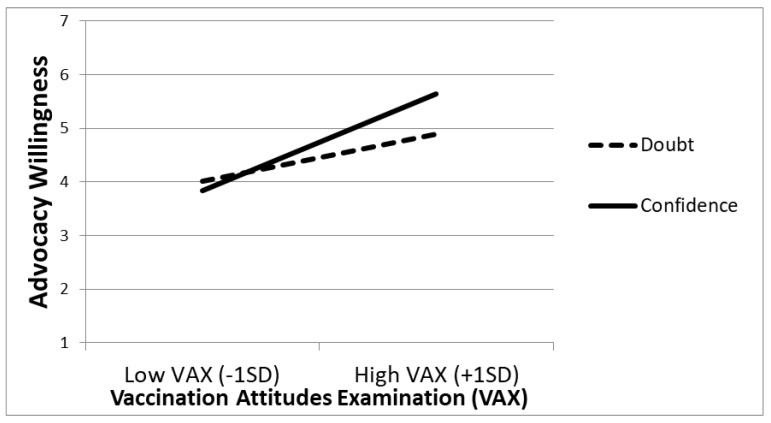
Study 2. Advocacy Willingness as a Function of VAX Responses and Manipulated Certainty.

## Data Availability

The data presented in this study are openly available in The Open Science Framework (OSF) at doi: 10.17605/OSF.IO/FPM3J.

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
