# Peer review of "Strengthening the Link between Vaccine Predispositions and Vaccine Advocacy through Certainty"

_vaccines, 2022, doi:10.3390/vaccines10111970_

Round 1

Reviewer 1 Report

The article entitled “Strengthening the link between vaccine predispositions and vaccine advocacy throuh certainty” is a research  that provides about the predictive validity of the Medicine Questionnaire (BMQ) and VAX scales that can predict important vaccination-related outcomes. The research was conducted using two studies. The first study has shown the link between a vaccination-related scale and advocary willingness and the second study has used the Vaccination Attitudes Examination (VAX) scale showing an association between with vaccinations, such as influenza vaccination and Covid-19, and intentions to receive future vaccines.  The topic is original and well organized and can provide useful interventions to involve people to have pro-vaccination behaviors. The article is susceptible of publication.

Author Response

We appreciate the kind and positive comments from reviewer 1

Reviewer 2 Report

Review Vaccines 2008334: Vaccine predispositions and advocacy

Abstract

No information on the population studied.

Introduction

Clear explanation of rationale for study, addressing important issue.

Line 93 BMQ – presumably Beliefs about Medicines Questionnaire (line 58) but abbreviation needs to be added after this first mention of the full name.

Line 99-100  – Measurement of ‘intention to advocate’ – would be useful to add some comment on how this intention can be measured and the validity of such measurements. (This is addressed in more detail in line 183-192 – but a brief summary at this point would assist the reader – including the information in lines 200-205, which strengthens the relevance of the measurement).

Study 1

Line 112 – 119 provides further information about advocacy, but does not discuss issues of measurement of ‘intention to advocate’.

Line 143 – This suggests that the questionnaire was administered through personal interview. The methods section does not clarify or describe how the questionnaire was administered.

Line 242 -250  - This section provides helpful information on the rationale for the second study. While this section follows on from the results of the first study, it is not necessary to know these results in order to understand the argument for the need for a further study. The authors could consider providing some of this information in the introduction to clarify the rationale for two studies.

Study 2

Line 308 – SVT ? would be useful to spell out this abbreviation again (although spelled out in line 70, I had forgotten it by this point in the paper)

Conclusions

While this section summarises the main findings and their implications, there is little consideration to factors that could possibly influence the findings. For example, the study was undertaken post the commencement of the covid 19 pandemic, in a context of considerable discussion and debate about the benefits and risk of vaccination. This heightened debate could potentially have increased the degree of uncertainty or anxiety in  regard to vaccination. It is also potentially relevant that the respondents were undergraduate university students, who are likely to be better educated than the general population, and more prepared to accept scientific advice on the benefits of vaccination.

Further research using different populations could also be recommended.

Author Response

We appreciate the positive and constructive feedback provided by reviewer 2. Below, we have organized our responses to his comments.

1. Abbreviations and clarifications. Reviewer 2 (R2) indicated that some abbreviations (BMQ, VAX, SVT) were not clearly introduced in the original manuscript. In addition, R2 also suggested explaining earlier in the introduction the rationale behind the choice of our dependent variables. Lastly, R2 also recommended highlighting in the introduction the presence of two studies (one with measured certainty and one with manipulated certainty) as well as where the studies were physically carried out.

We appreciate how these clarifications would contribute to improve the reading and understanding of the revised manuscript. Therefore, we have made the following changes in the article according to these suggestions made by R2.

We have incorporated the following phrase earlier in the introduction (line 100): “Given that COVID-19 vaccines were already widely available at the time of data collection, we decided to focus on another relevant outcome regarding vaccination (i.e., willingness to share messages)”.

Line 66: “Specifically, we aim at measuring certainty in an initial exploratory study to test the presence of such effect, and then we aim at manipulating certainty in a follow-up study to provide causal inference of such effect”.

Line 133: “The study was carried out in a laboratory room at the university where participants were asked to complete a survey in a computer using the Qualtrics software”.

2. Conclusions. Reviewer 2 suggested discussing the possible influence of contextual factors, such as the general uncertainty at the point of the pandemic in which Study 1 was carried out. Specifically, R2 wondered how this uncertainty could have affected our findings. In addition, R2 also recommended to discuss how the characteristics of our sample, composed by college students, might have affected our results (e.g., college students are more prepared to understand scientific advice).

We agree with reviewer 2 and therefore we have added the following paragraphs to t the revised version of the manuscript:

Line 404: “One additional aspect about this research is that the general relationship between attitudes and intentions in this particular topic might have been lower than usual, given the overall level of uncertainty at the time of the pandemic in which the studies were carried out.”

Line 411: “Specifically, our experimental sample was comprised of undergraduate university students, that were expected to have higher academic knowledge and more familiarization with the scientific field. In line with the Self-Validation Theory (Briñol & Petty, 2022), these moderating effects of certainty might take place specially for people who are involved enough in the topic to consider not only their attitudes about such topic, but the certainty with which they hold their attitudes. Other less-involved samples might have a more heuristic use of certainty. In line with multiple roles of variables in persuasion settings (Petty & Briñol, 2012), future research should study the extent to which certainty has different roles in predicting the association between attitudes and intentions as a function of involvement.”